# Preventing Model Collapse in Deep Canonical Correlation Analysis by Noise Regularization

## Abstract

Multi-View Representation Learning (MVRL) aims to learn a unified representation of an object from multi-view data. Deep Canonical Correlation Analysis (DCCA) and its variants share simple formulations and demonstrate state-of-the-art performance. However, with extensive experiments, we observe the issue of model collapse, *i.e.*, the performance of DCCA-based methods will drop drastically when training proceeds. The model collapse issue could significantly hinder the wide adoption of DCCA-based methods because it is challenging to decide when to early stop. To this end, we develop NR-DCCA, which is equipped with a novel noise regularization approach to prevent model collapse. Theoretical analysis shows that the full-rank property of the transformation is the key to preventing model collapse, and our noise regularization constrains the neural network to be "full-rank". A framework to construct synthetic data with different common and complementary information is also developed to compare MVRL methods comprehensively. The developed NR-DCCA outperforms baselines stably and consistently in both synthetic and real-world datasets, and the proposed noise regularization approach can also be generalized to other DCCA-based methods such as DGCCA.

**Keywords:** Multi-view representation learning; Canonical Correlation Analysis; Deep Canonical Correlation Analysis; Noise regularization; Model collapse

## 1 Introduction

In recent years, multi-view representation learning (MVRL) has emerged as a core technology for learning from multi-source data and providing readily useful representations to downstream tasks (Sun et al., 2023; Yan et al., 2021), and it has achieved tremendous success in various applications, such as video surveillance (Guo et al., 2015; Feichtenhofer et al., 2016; Deepak, K. et al., 2021), medical diagnosis (Wei et al., 2019; Xu et al., 2020) and social media (Srivastava & Salakhutdinov, 2012; Karpathy & Fei-Fei, 2015; Mao et al., 2014; Fan et al., 2020). Specifically, we collect multi-source data of the same object, and each data source can be regarded as one view of the object. For instance, an object can be described simultaneously through texts, videos, and audio, which contain both common and complementary information of the object (Yan et al., 2021; Zhang et al., 2019b; Hwang et al., 2021; Geng et al., 2021), and the MVRL aims to learn a unified representation of the object from the multi-view data.

The key challenge of MVRL is to learn the intricate relationships of different views. The Canonical Correlation Analysis (CCA), which is one of the early and representative methods for MVRL, transforms all the views into a unified space by maximizing their correlations (Hotelling, 1992; Horst, 1961; Hardoon et al., 2004; Lahat et al., 2015; Yan et al., 2023; Sun et al., 2023). Through correlation maximization, CCA can identify the common information between different views and extract them to form the representation of the object. On top of CCA, DCCA further utilizes the powerful deep neural networks(DNNs) to transform each view and then adopts CCA to maximize the correlation of the transformed views (Andrew et al., 2013). Indeed, there are quite a few variants of DCCA, such as DGCCA (Benton et al., 2017), DCCAE (Wang et al., 2015), DVCCA (Wang et al., 2016), DTCCA (Wong et al., 2021) and DCCA_GHA (Chapman et al., 2022).

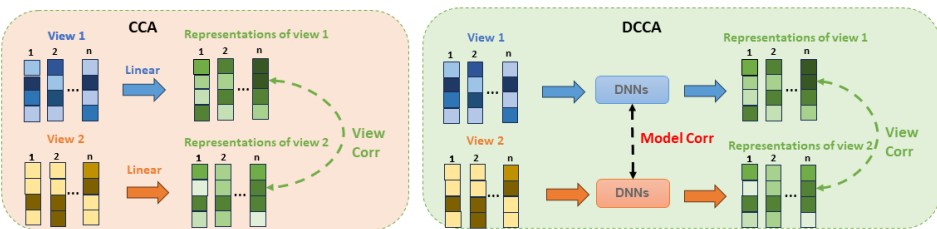

Figure 1: Correlations in CCA and DCCA.

However, with extensive experiments, **we observe that the DCCA-based methods generally perform well at the early training stage, while their performance drops drastically when the training proceeds.** This issue is referred to as model collapse. Essentially, the model collapse is mainly attributed to the overly powerful transformation abilities of DNNs, and the maximized view correlation may not come from views, but it may come from the model correlation among neural networks, as shown in Figure 1. Although representation produced by DCCA-based methods is naturally full-rank, the feature space may still become degenerated due to unregulated neural networks. In contrast, CCA-based methods utilize linear transformations, which are forced to be full-rank matrices, and hence there is no such model collapse issue.

Though early stopping could be adopted to prevent model collapse (Prechelt, 1998; Yao et al., 2007), it remains challenging when to stop. The model collapse issue of DCCA-based methods prevents the adoption in large models, and currently, many applications still use simple concatenation to combine different views (Yan et al., 2021; Zheng et al., 2020; Nie et al., 2017). Therefore, how to develop a DCCA-based MVRL method without model collapse remains an interesting and open question.

We prove that the main reason for CCA not having the model collapse issue is that the full-rank property holds in its transformation matrix, while the DNNs in DCCA do not possess such property. To this end, we develop a novel idea that employs noise regularization to enforce the DNNs to be "full-rank". Note that the proposed noise regularization approach is novel and particularly tailored for DCCA-based methods, which is different from the existing approaches that directly inject noise into the neural networks (Poole et al., 2014; He et al., 2019; Gong et al., 2020).

Overall, this paper develops NR-DCCA, a DCCA method equipped with a generalized noise regularization approach. The proposed noise regularization approach constrains the DNNs to be "full-rank", and it can also be applied to other DCCA-based methods. To justify the approach, the formulation of CCA is analyzed and rigorous proofs are provided. Comprehensive experiments using both synthetic datasets and real-world datasets demonstrate the consistent outperformance and stability of the developed NR-DCCA method. Our contributions are four-fold:

- The model collapse issue in DCCA-based methods for MVRL is identified, demonstrated, and explained.
- A simple yet effective noise regularization approach is proposed and the NR-DCCA method is developed to prevent model collapse.
- Rigorous proofs are provided to demonstrate that the full-rank property of the transformation in the CCA method is the key to preventing model collapse, which justifies the developed noise regularization approach from a theoretical perspective.
- A novel framework is proposed to construct synthetic data with different common and complementary information for comprehensively evaluating MVRL methods.

## 2 RELATED WORKS

### 2.1 MULTI-VIEW REPRESENTATION LEARNING

MVRL aims to uncover relationships among multi-view data in an unsupervised manner, thereby obtaining semantically rich representations that can be utilized for various downstream tasks (Sun

et al., 2023; Yan et al., 2021). A number of works have been proposed to deal with MVRL from different aspects. DMF-MVC (Zhao et al., 2017) utilizes deep matrix factorization to extract a shared representation from multiple views. MDcR (Zhang et al., 2016a) maps each view to a lower-dimensional space and applies kernel matching to enforce dependencies across the views. CPM-Nets (Zhang et al., 2019a) formalizes the concept of partial MVRL and many works have been proposed for such issue (Zhang et al., 2020; Tao et al., 2019; Li et al., 2022; Yin & Sun, 2021). $AE^2$-Nets (Zhang et al., 2019b) utilizes a two-level autoencoder framework to obtain a comprehensive representation of multi-view data. DUA-Nets (Geng et al., 2021) takes a generative modeling perspective and dynamically estimates the weights for different views. MVTCAE (Hwang et al., 2021) explores MVRL from an information-theoretic perspective, which can capture the shared and view-specific factors of variation by maximizing or minimizing specific total correlation. Our work focuses on CCA as a simple, classic, and theoretically sound approach as it can still achieve state-of-the-art performance consistently.

## 2.2 CCA AND ITS VARIANTS

Canonical Correlation Analysis (CCA) projects the multi-view data into a unified space by maximizing their correlations (Hotelling, 1992; Horst, 1961; Hardoon et al., 2004; Lahat et al., 2015; Yan et al., 2023; Sun et al., 2023). It has been widely applied in various scenarios that involve multi-view data, including dimension reduction (Zhang et al., 2016b; Sun et al., 2010a; Avron et al., 2013), classification (Kim et al., 2007; Sun et al., 2010b), and clustering (Fern et al., 2005; Chang & Lin, 2011). To further enhance the nonlinear transformability of CCA, Kernel CCA (KCCA) uses kernel methods, while Deep CCA (DCCA) employs DNNs. Since DNNs is parametric and can take advantage of large amounts of data for training, numerous DCCA-based methods have been proposed. Benton et al. (2017) utilizes DNNs to optimize the objective of Generalized CCA, with the aim of revealing connections between multiple views more effectively. To better preserve view-specific information, Wang et al. (2015) introduces the reconstruction errors of autoencoders to DCCA. Going a step further, Wang et al. (2016) proposes Variational CCA and utilizes dropout and private autoencoders to project common and view-specific information into two distinct spaces. Furthermore, there are many studies exploring efficient methods for computing the correlations between multi-view data when dealing with more than two views such as MCCA, GCCA and TCCA (Horst, 1961; Nielsen, 2002; Kettenring, 1971; Hwang et al., 2021). Some research focuses on improving the efficiency of computing CCA by avoiding the need for singular value decomposition (SVD) (Chang et al., 2018; Chapman et al., 2022). However, the model collapse issue of DCCA-based methods has not been explored and addressed.

## 2.3 NOISE REGULARIZATION

Noise regularization is a pluggable approach to regularize the neural networks during training (Bishop, 1995; An, 1996; Sietsma & Dow, 1991; Gong et al., 2020). In supervised tasks, Sietsma & Dow (1991) might be the first to propose that, by adding noise to the train data, the model will generalize well on new unseen data. Moreover, Bishop (1995); Gong et al. (2020) analyze the mechanism of the noise regularization and He et al. (2019); Gong et al. (2020) indicate that noise regularization can also be used for adversarial training to improve the generalization of the network. In unsupervised tasks, Poole et al. (2014) systematically explores the role of noise injection at different layers in autoencoders, and distinct positions of noise perform specific regularization tasks. However, how to make use of the noise regularization for DCCA-based methods, especially for preventing model collapse, has not been studied.

## 3 PRELIMINARIES

In this section, we will explain the objectives of the MVRL and then introduce CCA and DCCA as representatives of the CCA-based methods and DCCA-based methods, respectively. Lastly, the model collapse issue in DCCA-based methods is demonstrated.

## 3.1 SETTINGS FOR MVRL

Suppose the set of datasets from $K$ different sources that describe the same object is represented by $X$, and we define $X = \{X_1, \cdots, X_k, \cdots, X_K\}, X_k \in \mathbb{R}^{d_k \times n}$, where $x_k$ represents the $k$-th view ($k$-th data source), $n$ is the sample size, and $d_k$ represents the feature dimension for the $k$-th view. And we use $X_k'$ to denote the transpose of $X_k$.

The objective of MVRL is to learn a transformation function $\Psi$ that projects the multi-view data $X$ to a unified representation $Z \in \mathbb{R}^{m \times n}$, where $m$ represents the dimension of the representation space, as shown below:

$$Z = \Psi(X) = \Psi(X_1, \cdots, X_k, \cdots, X_K). \tag{1}$$

After applying $\Psi$ for representation learning, we expect that the performance of using $Z$ would be better than directly using $X$ for various downstream tasks.

## 3.2 CANONICAL CORRELATION ANALYSIS

Among various MVRL methods, CCA projects the multi-view data into a common space by maximizing their correlations. We first define the correlation between two views as follows:

$$\text{Corr}(W_1 X_1, W_2 X_2) = \text{tr}((\Sigma_{11}^{-1/2} \Sigma_{12} \Sigma_{22}^{-1/2})' \Sigma_{11}^{-1/2} \Sigma_{12} \Sigma_{22}^{-1/2})^{1/2} \tag{2}$$

where tr denotes the matrix trace, $\Sigma_{11}, \Sigma_{22}$ represent the self-covariance matrices of the projected views, and $\Sigma_{12}$ is the cross-covariance matrix between projected views (D'Agostini, 1994; Andrew et al., 2013). The correlation between the two projected views can be regarded as the sum of all singular values of the normalized cross-covariance (Hotelling, 1992; Anderson et al., 1958).

For multiple views, their correlation is defined as the summation of all the pairwise correlations (Nielsen, 2002; Kettenring, 1971), which is shown as follows:

$$\text{Corr}(W_1 X_1, \cdots, W_k X_k, \cdots, W_K X_K) = \sum_{k<j} \text{Corr}(W_k X_k, W_j X_j). \tag{3}$$

Essentially, CCA searches for the linear transformation matrices $\{W_k\}_k$ that maximize correlation among all the views. Mathematically, it can be represented as:

$$\{W_k^*\}_k = \arg \max_{\{W_k\}_k} \text{Corr}(W_1 X_1, \cdots, W_k X_k, \cdots, W_K X_K). \tag{4}$$

Once $W_k^*$ is obtained, the multi-view data are projected into a unified space. Lastly, all projected data are concatenated to obtain $Z = [W_1^* X_1; \cdots; W_k^* X_k; \cdots; W_K^* X_K]$ for downstream tasks.

## 3.3 DCCA

On top of the CCA, DCCA integrates neural networks and CCA to capture the nonlinear relationship among multi-view data. The major difference of DCCA is that the transformation matrix $W_k$ is replaced with multi-layer perceptrons (MLP), and the parameters in MLP are updated through backpropagation (Andrew et al., 2013). Specifically, each $W_k$ is replaced with a neural network $f_k$, which can be viewed as a nonlinear transformation. Similar to CCA, the goal of DCCA is to solve the following optimization problem:

$$\{f_k^*\}_k = \arg \max_{\{f_k\}_k} \text{Corr}\left(f_1(X_1), \cdots, f_k(X_k), \cdots, f_K(X_K)\right). \tag{5}$$

Again, the unified representation is obtained by $Z = [f_1^*(X_1); \cdots; f_k^*(X_k); \cdots; f_K^*(X_K)]$ for downstream tasks.

## 3.4 MODEL COLLAPSE OF DCCA

Despite exhibiting promising performance, DCCA shows a significant decline in performance as the training proceeds. We attribute this decline-in-performance phenomenon as model collapse. Model collapse could make the performance of DCCA-based methods even worse than classic CCA

methods and simple feature concatenation. As illustrated in (a) of Figure 4, one can see that the performance of DCCA drops drastically during training, though its best performance at early iterations could exceed that of CCA. The correlation between unrelated data increases as the collapsed model transforms any data to a degenerated feature space. In (b) of Figure 4, we can see that except for the proposed method (NR-DCCA), the correlation between unrelated data increases significantly. We could also analyze the model collapse in real-world data by feature visualization as shown in Figure 9 in Appendix A.9. However, it should be noted that the full-rank property of representation always holds for DCCA (verified in Appendix A.6.1), while this cannot prevent the model collapse. Indeed, the model collapse of DCCA is mainly due to the difference between $W_k$ and $f_k$, which requires further investigation.

## 4 DCCA WITH NOISE REGULARIZATION(NR-DCCA)

### 4.1 METHOD

In this section, we present NR-DCCA, which makes use of the noise regularization approach to prevent model collapse in DCCA. Indeed, the developed noise regularization approach can be applied to variants of DCCA methods, such as Deep Generalized CCA (DGCCA) (Benton et al., 2017). An overview of the NR-DCCA framework is presented in Figure 2.

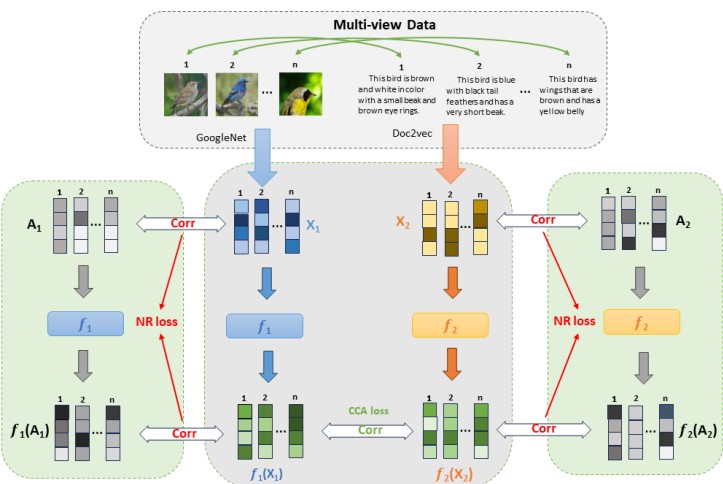

Figure 2: Illustration of NR-DCCA. We take the CUB dataset as an example: similar to DCCA, the $k$-th view $X_k$ is transformed using $f_k$ to obtain new representation $f_k(X_k)$ and then maximize the correlation between new representations. Additionally, for the $k$-th view, we incorporate the proposed NR loss to regularize $f_k$.

The key idea in NR-DCCA is to generate a set of i.i.d Gaussian white noise, denoted as $A = \{A_1, \cdots, A_k, \cdots, A_K\}, A_k \in \mathbb{R}^{d_k \times n}$, with the same shape as the multi-view data $X_k$. In CCA, the correlation with noise is invariant to the full-rank linear transformation $W_k$: $\text{Corr}(X_k, A_k) = \text{Corr}(W_k X_k, W_k A_k)$ (rigorous proof provided in Proposition 3). However, for DCCA, $\text{Corr}(X_k, A_k)$ might not equal $\text{Corr}(f_k(X_k), f_k(A_k))$ because the powerful neural networks $f_k$ have overfitted to the maximization problem in DCCA and "created" correlation itself. Therefore, we enforce the DCCA to mimic the behavior of CCA by adding a NR loss $\zeta_k = |Corr(f_k(X_k), f_k(A_k)) - Corr(X_k, A_k)|$, and hence the formulation of NR-DCCA is:

$$\{f_k^*\}_k = \arg \max_{\{f_k\}_k} \text{Corr}\left(f_1(X_1), \cdots, f_K(X_K)\right) - \alpha \sum_{k=1}^{K} \zeta_k. \tag{6}$$

where $\alpha$ is the hyper-parameter weighing the NR loss. NR-DCCA can be trained through backpropagation with the randomly generated $A$ in each epoch, and the unified representation is obtained directly using $\{f_k^*\}_k$ in the same manner as DCCA.

## 4.2 THEORETICAL ANALYSIS

In this section, we provide the rationale for why the developed noise regularization can help to prevent model collapse. Without loss of generality, we assume that all the datasets $X_k$ are zero-centered with respect to row (Hotelling, 1992). Then we make use of the following proposition to simplify our analysis:

**Proposition 1** *Given a specific matrix $B$ and a zero-centered $C$ with respect to rows, the product $BC$ is also zero-centered with respect to rows.*

This implies that $W_k A_k$ and $W_k X_k$ are both zero-centered matrices. When computing the covariance matrix, there is no need for an additional subtraction of the mean of row, which simplifies our subsequent derivations.

We first analyze CCA in detail, and the Moore-Penrose Inverse (MPI) (Petersen et al., 2008) will be used for analysis, and the MPI is defines as follows:

**Definition 1** *Given a specific matrix $Y$, its Moore-Penrose Inverse (MPI) is denoted as $Y^+$. $Y^+$ satisfies: $YY^+Y = Y$, $Y^+YY^+ = Y^+$, $YY^+$ is symmetric, and $Y^+Y$ is symmetric.*

The MPI $Y^+$ is unique and always exists for any $Y$. Furthermore, when matrix $Y$ is invertible, its inverse matrix $Y^-$ is exactly $Y^+$. Using the definition of MPI, we can rewrite the formulation of CCA. In particular, $\text{Corr}(\cdot, \cdot)$ can be derived by replacing the inverse with MPI. Using $\text{Corr}(X_k, A_k)$ as an example, the following proposition holds:

**Proposition 2 (MPI-based CCA)** *For the $k$-th view data $X_k$ and the Gaussian white noise $A_k$, we have*

$$Corr(X_k, A_k) = \frac{1}{(n-1)^2} tr(A_k^+ A_k X_k^+ X_k)^{1/2}, \forall k. \tag{7}$$

Utilizing the new form of $Corr$, we discover that the transformation matrix $W_k$ obtained from multi-view data $X = \{X_1, \cdots, X_k, \cdots, X_K\}, X_k \in \mathbb{R}^{d_k \times n}$ $(d_k < n)$ through CCA possesses the following properties:

**Proposition 3** *For any $k$, if $W_k$ is a square and full-rank matrix, the correlation between $X_k$ and $A_k$ remains unchanged before and after the transformation by $W_k$. Mathematically, we have $Corr(X_k, A_k) = Corr(W_k X_k, W_k A_k)$.*

**Proposition 4** *For any $k$, if $Corr(W_k X_k, W_k A_k) = Corr(X_k, A_k)$ and $W_k$ is a square matrix, then $W_k$ must be a full-rank matrix.*

Proofs of all the above propositions are presented in the Appendix. Combining both Proposition 3 and 4, we have Theorem 1 holds.

**Theorem 1 (Full-rank of CCA)** *If $W_k$ is a square matrix for any $k$, then $\eta_k = 0 \iff W_k$ is full-rank, where $\eta_k = |Corr(W_k X_k, W_k A_k) - Corr(X_k, A_k)|$.*

It is widely acknowledged that forced by the loss function, CCA searches for full-rank representation $W_k X_K$ and thereby obtains a full-rank matrix $W_k$ (refer to Lemma 2 in Appendix A.3). However, Theorem 1 is profound, as it first time provides the equivalent condition of the full-rank property of $W_k$ by connecting it to the correlation with noise, and then we can transplant this condition to DCCA. Essentially, CCA searches for $W_k$ as a full-rank matrix, and it is robust to random noise $A_k$, given the fact that the correlations between $X_k$ and $A_k$ will be invariant to transformation $W_k$. This indicates the $W_k$ will not learn to "create" correlation during the maximization process. However, the correlation increases in the DCCA training, which is closely related to the model collapse. We would like the DCCA to hold the same property as CCA, therefore, we define the "full-rank" property for the neural network $f_k$:

**Definition 2 ("Full-rank" of $f_k$)** *Given $k$, the neural network $f_k$ is called "full-rank" if $\zeta_k = 0$.*

Based on the Definition 2, the proposed NR-DCCA maximizes the correlation among views and also enforces the neural network to be "full-rank". As the neural network $f_k$ shares the same property of the full-rank matrix $W_k$, the model collapse issue can be eliminated. It should be noted that the "full-rank" property of $f_k$ is different from the full-rank of the representation $f_k(X_k)$, as the latter is defined by the matrix rank, while the former one is the new concept in this paper defined by $\zeta_k = 0$.

## 5 Numerical Experiments

We conduct extensive experiments on both synthetic and real-world datasets to answer the following research questions:

- **RQ1:** How can we construct synthetic datasets to evaluate the MVRL methods in a comprehensive manner?
- **RQ2:** Does NR-DCCA avoid model collapse across all synthetic MVRL datasets?
- **RQ3:** Does NR-DCCA perform consistently in real-world datasets?

We follow the protocol described in Hwang et al. (2021) for evaluating the MVRL methods. For each dataset, we construct a train dataset and a test dataset. All methods are trained in an unsupervised manner on the train dataset. Subsequently, the test dataset is used to obtain the representation, which will be evaluated in downstream tasks. For the regression task, we employ Ridge Regression (Hoerl & Kennard, 1970) and use $R2$ as the evaluation metric. For the classification task, we use Support Vector Classifier (SVC) (Chang & Lin, 2011) and report the average F1 scores. All tasks are evaluated using 5-fold cross-validation, and the reported results correspond to the average values of the respective metrics. Baseline methods include **CONCAT**, **CCA** (Hotelling, 1992), **PRCCA** (Tuzhilina et al., 2023), **KCCA** (Akaho, 2006), **DCCA** (Andrew et al., 2013), **DGCCA** (Benton et al., 2017), **DCCAE/DGCCAE** (Wang et al., 2015), **DCCA_PRIVATE/DGCCA_PRIVATE** (Wang et al., 2016), and **MVTCAE** (Hwang et al., 2021). Details of the experiment settings including datasets and baselines are presented in Appendix A.5. Hyper-parameter settings, including ridge regularization and $\alpha$ of NR, are discussed in Appendix A.6. We also analyze the computational complexity of different DCCA-based methods in Appendix A.8 and the learned representations are visualized in Appendix A.9. In the main paper, we mainly compare DCCA and NR-DCCA, while the results related to DGCCA are similar and presented in Appendix A.10.

### 5.1 Construction of Synthetic Datasets (RQ1)

We construct synthetic datasets to assess the performance of MVRL methods, and the framework is illustrated in Figure 3. We believe that the multi-view data describes the same object, which is represented by a high-dimensional embedding $G^{d \times n}$, where $d$ is the feature dimension and $n$ is the size of the data, and we call it God Embedding. Each view of data is regarded as a non-linear transformation of part (or all) of $G$. For example, we choose $K = 2, d = 100$, and then $X_1 = \phi_1(G[0 : 50 + \text{CR}/2, :]), X_2 = \phi_x(G[50 - \text{CR}/2 : 100], :)$, where $\phi_1$ and $\phi_2$ are non-linear transformations, and $CR$ is referred to as common rate. The common rate is defined as follows:

**Definition 3 (Common Rate)** *For two view data* $X = \{X_1, X_2\}$, *common rate is defined as the percentage overlap of the features in* $X_1$ *and* $X_2$ *that originate from* $G$.

One can see that the common rate ranges from $0\%$ to $100\%$. The larger the value, the greater the correlation between the two views, and a value of $0$ indicates that the two views do not share any common dimensions in $G$. Additionally, we construct the downstream tasks by directly transforming the God Embedding $G$. Each task $T_j = \psi_j(G)$, where $\psi_j$ is a transformation, and $T_j$ represents the $j$-th task. By setting different $G$, common rates, $\phi_k$, and $\psi_j$, we can create various synthetic datasets to evaluate the MVRL methods. Finally, $X_k$ are observable to the MVRL methods for learning the representation, and the learned representation will be used to classify/regress $T_j$ to examine the performance of each method. Detailed implementation is given in Appendix A.7.

### 5.2 Performance on synthetic datasets (RQ2)

We generate the synthetic datasets with different common rates, and the proposed NR-DCCA and other baseline methods are compared, as presented in (a) of Figure 4. The values depicted in the Figure represent the mean and standard deviation of the performance of methods across datasets.

One can see that the DCCA-based methods (*e.g.*, DCCA, DCCAE, DCCA_PRIVATE) will encounter model collapse during the training, and the variance of accuracy also increases. CCA-based methods (CCA and KCCA) demonstrate a stable performance, while the best accuracy is not as good as DCCA-based methods. Our proposed NR-DCCA achieves the state-of-the-art performance as well as training stability to prevent model collapse.

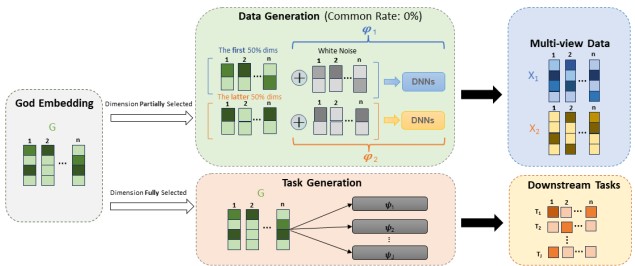

Figure 3: Construction of a synthetic dataset. This example consists of 2 views and $n$ objects, and the common rate is $0\%$.

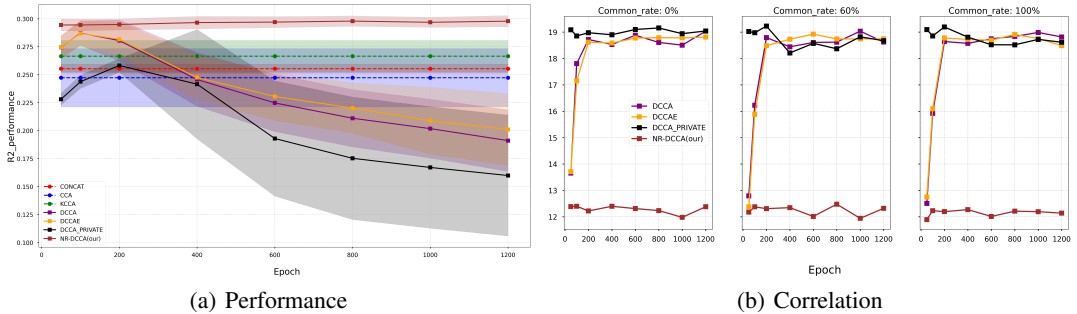

(a) Performance            (b) Correlation

Figure 4: (a) Mean and standard deviation of the CCA-based method performance across synthetic datasets in different training epochs. (b) The correlation between noise and real data after transformation varies with epochs in different common rate settings for CCA-based methods.

Moreover, according to our analysis, the correlation should be invariant if neural networks hold the "full-rank" property. So, after training DCCA, DGCCA, and their NR-variants, we utilize the trained encoders to project the corresponding view data and randomly generated Gaussian white noise and then compute their correlation, as shown in (b) of Figure 4. It can be observed that except for our method (NR-DCCA), as training progresses, other methods increase the correlation between unrelated data. It should be noted that this phenomenon always occurs under any common rates.

The performance under different comment rates is inspected separately, as presented in Figure 5. The results at the final epoch are also presented in Table 2 in Appendix A.11. Overall, all the methods demonstrate an improving performance when the common rate increases. DCCA-based methods could achieve similar performance as NR-DCCA, while they collapse drastically when the training proceeds. The NR-DCCA has a consistent outperformance over other methods during the training.

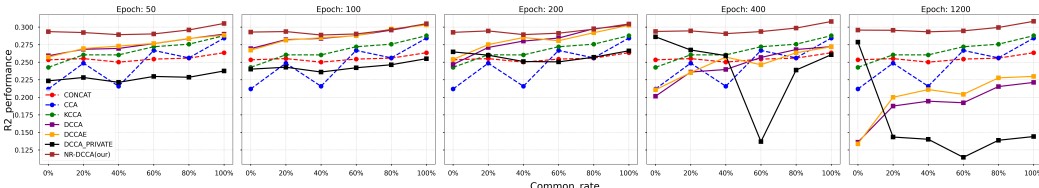

Figure 5: Performance of different methods with respect to data with different common rates during the training. Each column represents the testing accuracy of the method at a specific training epoch.

An interesting observation is that, most CCA-based methods perform better on data with high common rates, consistent with CCA's assumption of maximizing the correlation between data from different views. However, this also indicates that their performance may degrade with low correlations among views.

### 5.3 CONSISTENT PERFORMANCE ON REAL-WORLD DATASETS (RQ3)

We further conduct experiments on three real-world datasets: **PolyMnist** (Sutter et al., 2021), **CUB** (Wah et al., 2011), **Caltech** (Deng et al., 2018). Additionally, we use different number of views in **PolyMnist**. The results are presented in Figure 6, and the performance of the final epoch in the figure is presented in Table 2 in the Appendix A.11. Generally, the proposed NR-DCCA demonstrates a competitive and stable performance. Different from the synthetic data, the performance of CCA-based methods under-perform the DCCA-based methods, which might be due to the complex nature of the real-world views. However, the DCCA-based methods still exhibit varying degrees of collapse as the number of epochs increases. It is noteworthy that on the PolyMnist dataset, as the number of views increases, model collapses become more severe for DCCA-based methods, while our NR-DCCA can further improve the performance.

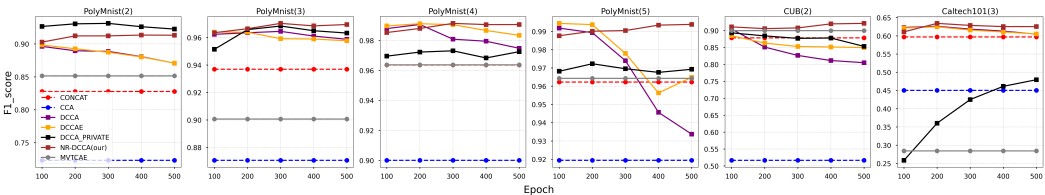

Figure 6: Performance of different methods in real-world datasets. Each column represents the performance on a specific dataset. The number of views in the dataset is denoted in the parentheses next to the dataset name.

## 6 CONCLUSIONS

We propose a novel noise regularization approach for DCCA in the context of MVRL, and it can prevent model collapse during the training, which is an issue observed and analyzed in this paper for the first time. Specifically, we theoretically analyze the full-rank property in CCA and demonstrate that it is the key to preventing model collapse. To this end, an analogy of the "full-rank" property is defined for DCCA, and the noise regularization can be considered as an implicit constraint to make the neural networks "full-rank". Additionally, synthetic datasets with different common rates are generated and tested on, which provide a benchmark for fair and comprehensive comparisons of different MVRL methods. The NR-DCCA developed in the paper inherits the merits of both CCA and DCCA to achieve stable and consistent outperformance in both synthetic and real-world datasets. More importantly, the proposed noise regularization approach can also be generalized to other DCCA-based methods (*e.g.*, DGCCA).

In future studies, we wish to explore the potential of noise regularization in other representation learning tasks, such as contrastive learning and generative models. It is also interesting to further investigate the "full-rank" definition of neural networks, and how it is different from other neural network regularization approaches, such as orthogonalization (Bansal et al., 2018; Huang et al., 2020) and weight decay (Loshchilov & Hutter, 2017; Zhang et al., 2018; Krogh & Hertz, 1991). Additionally, it is also interesting but challenging to investigate what (e.g., parameters, gradients, features) and how the neural network has been regularized through the noise. Our ultimate goal is to make the developed noise regularization a pluggable and useful module for neural network regularization.

## REPRODUCIBILITY STATEMENT

All of our experiments are conducted with fixed random seeds and all the performance of down-stream tasks is the average value of a 5-fold cross-validation. The CCA-zoo package is adopted as the implementation of various CCA/GCCA-based methods, and the original implementation of MVTCAE is employed. Both baselines and our developed NR-DCCA/NR-DGCCA are implemented in the same PyTorch environment (see requirements.txt in the source codes). All the datasets used in the paper are either provided or open datasets. Detailed proofs of all the propositions in the main paper can be found in the Appendix. Both source codes and appendix can be downloaded from the supplementary material.

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

# A APPENDIX

## A.1 PROOF OF PROPOSITION 1

Let $B_{i,j}$ and $C_{i,j}$ denote the $(i,j)$-th entry of $B$ and $C$, respectively. Then we have:

$$(BC)_{i,j} = \sum_{r=1} B_{i,r} C_{r,j}$$

Since each row of $C$ has a mean of 0, we have $\sum_{j=1}^{n} C_{r,j} = 0, \forall r$. For the mean value of $i$-th row of $BC$, we can write:

$$
\begin{aligned}
\frac{1}{n} \sum_{j=1}^{n} (BC)_{i,j} &= \frac{1}{n} \sum_{j=1}^{n} \sum_{r=1} B_{i,r} C_{r,j} \\
&= \frac{1}{n} \sum_{r=1} \sum_{j=1}^{n} B_{i,r} C_{r,j} \\
&= \frac{1}{n} \sum_{r=1} B_{i,r} \left( \sum_{j=1}^{n} C_{r,j} \right) \\
&= \frac{1}{n} \sum_{r=1} B_{i,r} \cdot 0 \\
&= 0
\end{aligned}
\tag{8}
$$

## A.2 PROOF OF PROPOSITION 2

$$
\begin{aligned}
\text{Corr}(X_k, A_k) &= \text{tr}((\Sigma_{11}^{-1/2} \Sigma_{12} \Sigma_{22}^{-1/2})' \Sigma_{11}^{-1/2} \Sigma_{12} \Sigma_{22}^{-1/2})^{1/2} \\
&= \text{tr}(\Sigma_{22}^{-1/2} \Sigma_{12}' \Sigma_{11}^{-1/2} \Sigma_{11}^{-1/2} \Sigma_{12} \Sigma_{22}^{-1/2})^{1/2} \\
&= \text{tr}(\Sigma_{22}^{-1/2} \Sigma_{22}^{-1/2} \Sigma_{12}' \Sigma_{11}^{-1/2} \Sigma_{11}^{-1/2} \Sigma_{12})^{1/2} \\
&= \text{tr}(\Sigma_{22}^{-1} \Sigma_{12}' \Sigma_{11}^{-1} \Sigma_{12})^{1/2} \\
&= \frac{1}{(n-1)^2} \text{tr}((A_k A_k')^{-1} (X_k A_k')' (X_k X_k')^{-1} (X_k A_k'))^{1/2} \\
&= \frac{1}{(n-1)^2} \text{tr}((A_k X_k')^{-1} (A_k X_k') (X_k X_k')^{-1} (X_k A_k'))^{1/2} \\
&= \frac{1}{(n-1)^2} \text{tr}((A_k A_k')^{+} (A_k X_k') (X_k X_k')^{+} (X_k A_k'))^{1/2} \\
&= \frac{1}{(n-1)^2} \text{tr}(A_k' (A_k A_k')^{+} A_k X_k' (X_k X_k')^{+} X_k)^{1/2} \\
&= \frac{1}{(n-1)^2} \text{tr}(A_k^{+} A_k X_k^{+} X_k)^{1/2}
\end{aligned}
\tag{9}
$$

The first row is based on the definition of Corr, the second row is because trace is invariant under cyclic permutation, the fifth row is to replace matrix inverse by MPI and the ninth row is due to $Y^{+} = Y'(YY^{+})$ (Petersen et al., 2008).

## A.3 PROOF OF PROPOSITION 3

Firstly, we have the $k$-th view data $X_k$ to be full-rank, as we can always delete the redundant data, and the random noise $A_k$ is full-rank as each column is generated independently. Then by utilizing Proposition 2, we derive that the correlation between $X_k$ and $A_k$ remains unchanged before and after the transformation:

$$
\begin{aligned}
\text{Corr}(W_k X_k, W_k A_k) &= \frac{1}{(n-1)^2} \text{tr}((W_k A_k)^+ W_k A_k (W_k X_k)^+ W_k X_k)^{1/2} \\
&= \frac{1}{(n-1)^2} \text{tr}((W_k^+ W_k A_k)^+ (W_k A_k A_k^+)^+ W_k A_k (W_k^+ W_k X_k)^+ (W_k X_k X_k^+)^+ W_k X_k)^{1/2} \\
&= \frac{1}{(n-1)^2} \text{tr}((W_k^+ W_k A_k)^+ W_k^+ W_k A_k (W_k^+ W_k X_k)^+ W_k^+ W_k X_k)^{1/2} \\
&= \frac{1}{(n-1)^2} \text{tr}(A_k^+ A_k X_k^+ X_k)^{1/2} \\
&= \text{Corr}(X_k, A_k)
\end{aligned}
\tag{10}
$$

The first row is based on Proposition 2, the second row is because given two matrices $B$ and $C$, $(BC)^+ = (B^+ BC)^+ (BC^+ C)^+$ always holds (Petersen et al., 2008), and the third row utilizes the properties of full-rank and square matrix $W_k$: $W_k^+ = W_k^-$, which means $W_k^+ W_k = W_k W_k^+ = I_{d_k}$ (Petersen et al., 2008).

## A.4 PROOF OF PROPOSITION 4

This proposition is equivalent to its contra-positive proposition: if $W_k$ is not a full-rank matrix, there exists random noise data $A_k$ such that $\eta_k = |Corr(W_k X_k, W_k(A_k)) - Corr(X_k, A_k)|$ is not 0. And we find that when $W_k$ is not full-rank, there exists $A_k = X_k$ such that $\eta_k \neq 0$. We have the following derivation:

$$
\begin{aligned}
\eta_k &= |Corr(W_k X_k, W_k A_k) - Corr(X_k, A_k)| \\
&= \left| \frac{1}{(n-1)^2} \text{tr}((W_k A_k)^+ (W_k A_k)(W_k X_k)^+ (W X_k))^{1/2} - \frac{1}{(n-1)^2} \text{tr}(A^+ A X^+ X)^{1/2} \right| \\
&= \left| \frac{1}{(n-1)^2} \text{tr}((W_k^+ W_k A_k)^+ (W_k A_k A_k^+)^+ W_k A_k (W_k^+ W_k X_k)^+ (W_k X_k X_k^+)^+ W_k X_k)^{1/2} - \frac{1}{(n-1)^2} \text{tr}(A_k^+ A_k X_k^+ X_k)^{1/2} \right| \\
&= \left| \frac{1}{(n-1)^2} \text{tr}((W_k^+ W_k A_k)^+ W_k^+ W_k A_k (W_k^+ W_k X_k)^+ W_k^+ W_k X_k)^{1/2} - \frac{1}{(n-1)^2} \text{tr}(A_k^+ A X_k^+ X_k)^{1/2} \right|
\end{aligned}
\tag{11}
$$

The first row is the definition of NR loss with respect to $W_k$, the second row is based on the new form of CCA, the third row is because given two specific matrices $B$ and $C$, it holds the equality $(BC)^+ = (B^+ BC)^+ (BC^+ C)^+$ (Petersen et al., 2008), and the fourth row utilizes the properties of full-rank matrix: for full-rank matrices $X_k$ and $A_k$, whose sample size is larger than dimension size, they fulfill: $X_k X_k^+ = I_{d_k}, A_k A_k^+ = I_{d_k}$ (given a specific full-rank matrix $Y$, if its number of rows is smaller than that of cols, it holds that $Y^+ = Y'(YY')^-$, which means that $YY^+ = I$) (Petersen et al., 2008).

Let us analyze the case when $A_k = X_k$:

$$
\begin{aligned}
\eta_k &= \left| \frac{1}{(n-1)^2} \text{tr}((W_k^+ W_k X_k)^+ W_k^+ W_k X_k (W_k^+ W_k X_k)^+ W_k^+ W_k X_k)^{1/2} - \frac{1}{(n-1)^2} \text{tr}(X_k^+ X_k X_k^+ X_k)^{1/2} \right| \\
&= \left| \frac{1}{(n-1)^2} \text{tr}((W_k^+ W_k X_k)^+ W_k^+ W_k X_k)^{1/2} - \frac{1}{(n-1)^2} \text{tr}(X_k^+ X_k)^{1/2} \right|.
\end{aligned}
\tag{12}
$$

The first row is to replace $A_k$ with $X_k$, the second row is because $X_k^+ X_k X_k^+ = X_k^+$ and $(W_k^+ W_k X_k)^+ W_k^+ W_k X_k (W_k^+ W_k X_k)^+ = W_k^+ W_k X_k$, which are based on the definition of MPI that given a specific matrix $Y$, $Y^+ Y Y^+ = Y^+$ (Petersen et al., 2008).

Next, we need to prove the following two lemmas:

**Lemma 1** *Given a specific matrix $Y$ and its MPI $Y^+$, let $Rank(Y)$ and $Rank(Y^+ Y)$ be the ranks of $Y$ and $Y^+ Y$, respectively. It is true that:*

$$Rank(Y) = Rank(Y^+ Y)$$

$$Rank(Y^+ Y) = tr(Y^+ Y)$$

**Proof** *Firstly, the column space of $Y^+ Y$ is a subspace of the column space of $Y$. Therefore, $Rank(Y^+ Y) \leq Rank(Y)$. On the other hand, according to the definition of MPI (Petersen et al.,*

2008), we know that $Y = Y(Y^+Y)$. Since the rank of a product of matrices is at most the minimum of the ranks of the individual matrices, we have $Rank(Y) \leq Rank(Y^+Y)$. Combining the two inequalities, we have $Rank(Y) = Rank(Y^+Y)$.

Furthermore, since $(Y^+Y)(Y^+Y) = Y^+Y$ (it holds that $Y^+ = Y^+YY^+$ according to the definition of MPI (Petersen et al., 2008)), $Y^+Y$ is an idempotent and symmetric matrix, and thus its eigenvalues must be 0 or 1. So the sum of its eigenvalues is exactly its rank. Considering matrix trace is the sum of eigenvalues of matrices, we have $Rank(Y^+Y) = tr(Y^+Y)$.

**Lemma 2** $Rank(W_kX_k) < Rank(X_k)$ , when $W_k$ is not a full-rank matrix and $X_k$ is a full-rank matrix.

**Proof** Since the rank of a product of matrices is at most the minimum of the ranks of the individual matrices, we have $Rank(W_kX_k) \leq min(Rank(W_k), Rank(X_k))$. Considering $X_k$ is full-rank, $Rank(X_k) = min(d_k, n)$ and then $Rank(W_kX_k) \leq min(Rank(W_k), Rank(X_k)) = min(Rank(W_k), min(d_k, n))$. Since $W_k$ is not full-rank, we have $Rank(W_k) < d_k$. In conclusion, $Rank(W_kX_k) < min(d_k, min(d_k, n))$ and then $Rank(W_kX_k) < d_k \leq Rank(X_k)$.

As a result, we can know that when the random noise data $A_k$ is exactly $X_k$ and $W_k$ is not full-rank, $\eta_k$ can not be zero:

$$
\begin{aligned}
\eta_k &= \left| \frac{1}{(n-1)^2} tr((W_k^+W_kX_k)^+W_k^+W_kX_k)^{1/2} - \frac{1}{(n-1)^2} tr(X_k^+X_k)^{1/2} \right| \\
&= \left| \frac{1}{(n-1)^2} Rank(W_k^+W_kX_k)^{1/2} - \frac{1}{(n-1)^2} Rank(X)^{1/2} \right| \\
&\neq \left| \frac{1}{(n-1)^2} Rank(X_k)^{1/2} - \frac{1}{(n-1)^2} Rank(X_k)^{1/2} \right| \\
&\neq 0
\end{aligned}
\tag{13}
$$

The first row is due to Equation 12, the second row is based on Lemma 1 that $tr((W_k^+W_kX_k)^+W_k^+W_kX_k) = Rank(W_k^+W_kX_k)$ and $tr(X_k^+X_k) = Rank(X)$, and the third row is because of Lemma 2.

Finally, we have if $\eta_k$ is always constrained to 0 for any $A_k$, then $W_k$ must be a full-rank matrix.

### A.5 DETAILS OF DATASETS AND BASELINES

**Synthetic datasets**:

We make 6 groups of multi-view data originating from the same $G \in \mathbb{R}^{d \times n}$ (we set $n = 4000, d = 100$). Each group consists of tuples with 2 views (2000 tuples for training and 2000 tuples for testing) and a distinct common rate. Common rates of these sets are from $\{0\%, 20\%, 40\%, 60\%, 80\%, 100\%\}$ and there are 50 downstream regression tasks. We report the mean and standard deviation of R2 score across all the tasks.

**Real-world datasets**:

**PolyMnist** (Sutter et al., 2021): A dataset consists of tuples with 5 different MNIST images ($60,000$ tuples for training and $10,000$ tuples for testing). Each image within a tuple possesses distinct backgrounds and writing styles, yet they share the same digit label. The background of each view is randomly cropped from an image and is not used in other views. Thus, the digit identity represents the common information, while the background and writing style serve as view-specific factors. The downstream task is the digit classification task. **CUB** (Wah et al., 2011): A dataset consists of tuples with deep visual features (1024-d) extracted by GOOGLENET and text features (300-d) obtained through DOC2VEC (Le & Mikolov, 2014) (480 tuples for training and 600 tuples for testing). This MVRL task utilizes the first 10 categories of birds in the original dataset and the downstream task is the bird classification task. **Caltech** (Deng et al., 2018): A dataset consists of tuples with traditional visual features extracted from images that belong to 101 object categories, including an additional background category (6400 tuples for training and 9144 tuples for testing). Following Yang et al. (2021), three features are used as views: a $1,984$-d HOG feature, a 512-d GIST feature, and a 928-d SIFT feature.

**Baselines**:

Direct method:

- **CONCAT** straightforwardly concatenates original features from different views.

CCA methods:

- **CCA** (Hotelling, 1992) maps multiple views' data into a common space that maximizes their correlation and concatenates the new representations of different views.
- **PRCCA** Tuzhilina et al. (2023) preserves the internal data structure by grouping high-dimensional data features while applying an l2 penalty to CCA,.

Kernel CCA Methods:

- **KCCA** (Akaho, 2006) employs CCA methods through positive-definite kernels.

DCCA-based methods:

- **DCCA** (Andrew et al., 2013) employs neural networks to individually project multiple sets of views, obtaining new representations that maximize the correlation between each pair of views.
- **DGCCA** (Benton et al., 2017) constructs a shared representation and maximizes the correlation between each view and the shared representation.
- **DCCAE/DGCCAE** (Wang et al., 2015) introduces reconstruction objectives to DCCA, which simultaneously optimize the canonical correlation between the learned representations and the reconstruction errors of the autoencoders.
- **DCCA_PRIVATE/DGCCA_PRIVATE** (Wang et al., 2016) incorporates dropout and private autoencoders, thus preserving both shared and view-specific information.

Information theory-based methods:

- **MVTCAE** (Hwang et al., 2021) maximizes the reduction in Total Correlation to capture both shared and view-specific factors of variations.

All CCA-based methods leverage the implementation of CCA-Zoo (Chapman & Wang, 2021). To ensure fairness, we use the official implementation of MVTCAE while replacing the strong CNN backbone with MLP.

### A.6 HYPER-PARAMETER SETTINGS

To ensure a fair comparison, we tune the hyper-parameters of all baselines within the ranges suggested in the original papers, including hyper-parameter $r$ of ridge regularization, except for the following fixed settings:

The embedding size for the real-world datasets is set as 200, while the size for synthetic datasets is set as 100. Batch size is $\min(2000, \text{full-size})$. The same MLP architectures are used for DCCA-based methods.

In the synthetic datasets, DCCA, DGCCA, DCCAE, and DGCCAE methods utilize a minimum learning rate of $5e-3$. DCCA_PRIVATE/DGCCA_PRIVATE employ a slightly higher learning rate of $1e-2$. In contrast, our proposed methods, NR-DCCA/NR-DGCCA, utilize the maximum learning rate of $1.5e-2$. And the regularization weight $\alpha$ is set as 200.

In the real-world datasets, both the learning rates in PolyMnist and CUB are set to $1e-4$. For Caltech101, a slightly lower learning rate of $5e-5$ is used. To expedite the computation of $\text{Corr}(X_k, A_k)$, on the PolyMnist dataset, we utilize the initialized $f_k$ to reduce the feature dimensions of $X_k$ and $A_k$ separately. Subsequently, we calculate their correlation. For the extracted features in the CUB and Caltech101 datasets, we simply employ $X_k[:outdim,:]$ and $A_k[:outdim,:]$ to compute of Corr. The hyper-parameter $r$ of ridge regularization is set as 0 in our NR-DCCA and NR-DGCCA. The optimal $\alpha$ values of NR-DCCA for the CUB, PolyMnist, and Caltech datasets are found to be 1.5, 5, and 15, respectively.

### A.6.1 HYPER-PARAMETER $r$ IN RIDGE REGULARIZATION

In this section, we discuss the effects of hyper-parameter $r$ in ridge regularization. Ridge regularization is commonly used across almost all (D)CCA methods, whichimproves numerical stability. It works by adding an identity matrix $I$ to the estimated covariance matrix. However, ridge regularization mainly regularizes the features, rather than the transformation (i.e., $W_k$ in CCA and $f_k$ in DCCA) and it cannot prevent the neural networks from degenerating (i.e., model collapse). To further support our arguments, we provide the experimental results with different ridge parameters on a real-world dataset CUB as shown in Figure 7. One can see that the ridge regularization even damages the performance of DCCA. In our NR-DCCA, we actually set the ridge parameter to zero. We conjecture the reason is that the large ridge parameter could make the neural network even "lazier" to actively project the data into a better feature space, as the full-rank property of features and covariance matrix are already guaranteed, and this is also evidenced by the "Square sum of feature covariance" shown in the figure.

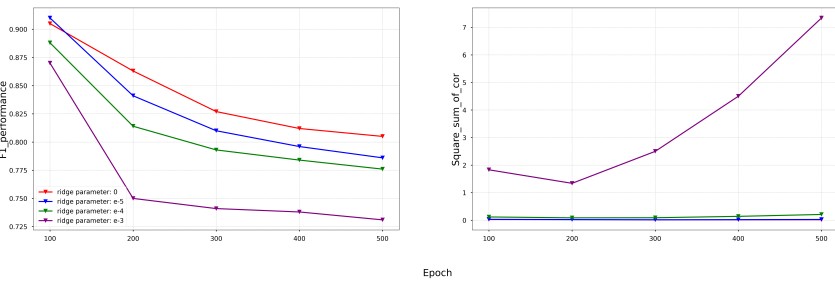

Figure 7: The effects of hyper-parameter $r$ of DCCA in the CUB dataset.

### A.6.2 HYPER-PARAMETER $\alpha$ OF NR-DCCA

The choice of the hyper-parameter $\alpha$ is essential in NR-DCCA. Different from the conventional hyper-parameter tuning procedures, the determination of $\alpha$ is simpler, as we can search for the smallest $\alpha$ that can prevent the model collapse, and the model collapse can be directly observed on the validation data. Specifically, we increase the $\alpha$ adaptively until the model collapse issue is tackled, i.e., the correlation with noise will not increase or the performance of DCCA will not drop with increasing training epochs, then the optimal $\alpha$ is found. To further illustrate the influence of $\alpha$ in NR-DCCA, we present performance curves of NR-DCCA in CUB under different $\alpha$. As shown in Figure 8, if $\alpha$ is too large, the convergence of the training becomes slow; if $\alpha$ is too small, model collapse still remains. Additionally, one can see the NR-DCCA outperforms DCCA robustly with a wide range of $\alpha$.

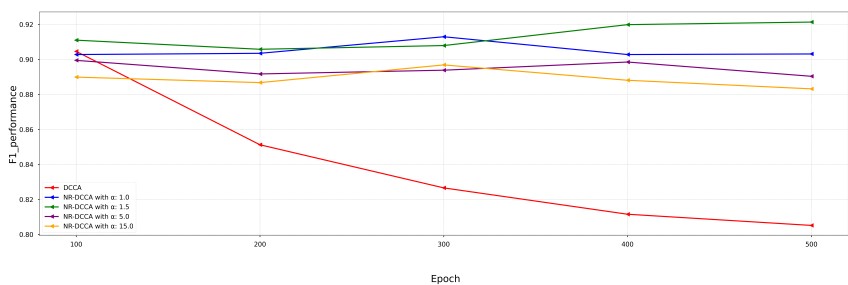

Figure 8: The effects of hyper-parameter $\alpha$ of NR-DCCA in the CUB dataset.

### A.7 IMPLEMENTATION DETAILS OF SYNTHETIC DATASETS

We draw $n$ random samples with dim $d$ from a Gaussian distribution as $G \in \mathbb{R}^{d \times n}$ to represent complete representations of $n$ objects. We define the non-linear transformation $\phi_k$ as the addition of noise to the data, followed by passing it through a randomly generated MLP. To generate the data

for the $k$-th view, we select specific feature dimensions from $G$ based on a given common rate 3 and then apply $\phi_k$ to those selected dimensions. And we define $\psi_j$ as a linear layer, and task $T_j$ is generated by directly passing G through $\psi_j$.

## A.8 COMPLEXITY ANALYSIS

In this section, we compare the computational complexity of different DCCA-based methods. Assuming that we have data from $K$ views, with each view containing $N$ samples and $D$ feature dimensions, then we have the computational complexity of each method in Table 1.

Table 1: Comparisons of computational complexity against baselines

|  | DCCA | DCCAE | DCCA_PRIVATE | NR-DCCA |
|---|---|---|---|---|
| Generation of Noise | - | - | - | $O(K*N*D)$ |
| MLP Encoder | $O(K*N*L*H^2)$ | $O(K*N*L*H^2)$ | $O(2*K*N*L*H^2)$ | $O(2*K*N*L*H^2)$ |
| MLP Decoder | - | $O(K*N*L*H^2)$ | $O(K*N*L*H^2)$ | - |
| Reconstruction Loss | - | $O(K*N*D)$ | $O(K*N*D)$ | - |
| Correlation Maximization | $O((M*K)^3)$ | $O((M*K)^3)$ | $O((M*K)^3)$ | $O((M*K)^3)$ |
| Noise Regularization | - | - | - | $O(2*K*(M*K)^3)$ |

- **Complexity of MLP:** We will use DNNs with the same MLP structure, consisting of $L$ hidden layers, each with $H$ neurons. Therefore, the computational complexity of one pass of the data through the DNNs can be expressed as $O(N*(D*H + D*M + L*H^2))$. To simplify, we use $O(N*L*H^2)$.

- **Complexity of Corr:** During the process of calculating $Cor$ among $K$ views, three main computations are involved. The calculation complexity of the covariance is $O(N*(M*K)^2$. Second, the complexity of the inverse matrix and the eigenvalues are $O((M*K)^3$. As a result, the computational complexity of calculating $Cor$ can be considered as $O((M*K)^3)$.

- **Complexity of reconstruction loss:** The reconstruction loss, also known as the mean squared error (MSE) loss, has a complexity of $O(N*D)$.

## A.9 VISUALIZATION OF THE LEARNED REPRESENTATIONS

To further demonstrate the effectiveness of our method, we employ 2D-tSNE visualization to depict the learned representations of the CUB dataset (test set) under different methods. Each data point is colored based on its corresponding class, as illustrated in Figure 9. There are a total of 10 categories, with 60 data points in each category. A reasonable distribution of learned representations entails that data points belonging to the same class are grouped together in the same cluster, which is distinguishable from clusters representing other classes. Additionally, within each cluster, the data points should exhibit an appropriate level of dispersion, indicating that the data points within the same class can be differentiated rather than collapsing into a single point. This dispersion is indicative of the preservation of as many distinctive features of the data as possible.

From Figure. 9, we can observe that CCA, DCCA / DGCCA have all confused the data from different categories. Specifically, CCA completely scatter the data points as it cannot handle nonlinear relationships. By incorporating autoencoders, DCCAE / DGCCAE and DCCA_PRIVATE / DGCCA_PRIVATE have partially separated the data; however, they have not fully separated the green and orange categories. NR-DCCA / NR-DGCCA is the only method that successfully separates all categories.

It is worth noting that our approach not only separates the data into different clusters but also maintains dispersion within each cluster. Unlike DCCA_PRIVATE / DGCCA_PRIVATE, where the data points within a cluster form a strip-like distribution, our method ensures that the data points within each cluster remain appropriately scattered.

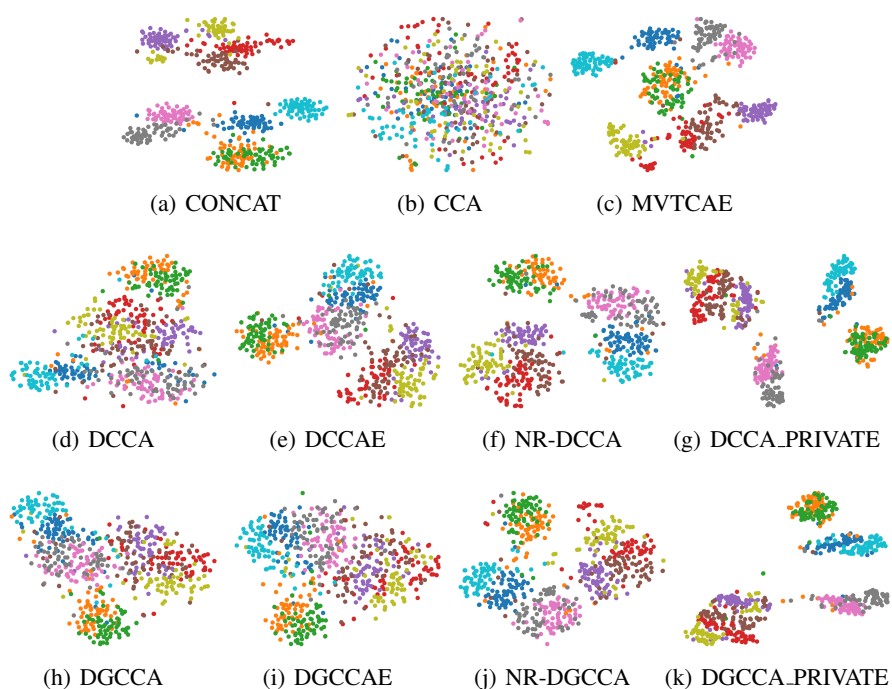

Figure 9: Visualization of the learned representations with t-SNE in the **CUB** dataset.

## A.10 DGCCA AND NR-DGCCA

This section presents the experimental results for DGCCA and NR-DGCCA, which supplement the results of GCCA and NR-DCCA presented in the main paper. In general, DGCCA is a variant of DCCA, and hence the proposed noise regularization approach can also be applied. We repeat the experiments in Figures 4, 5, and 6, and hence we have the results for DGCCA in Figure 10, 11, and 12. One can see that the proposed noise regularization approach can also help DGCCA prevent model collapse, proving its generalizability.

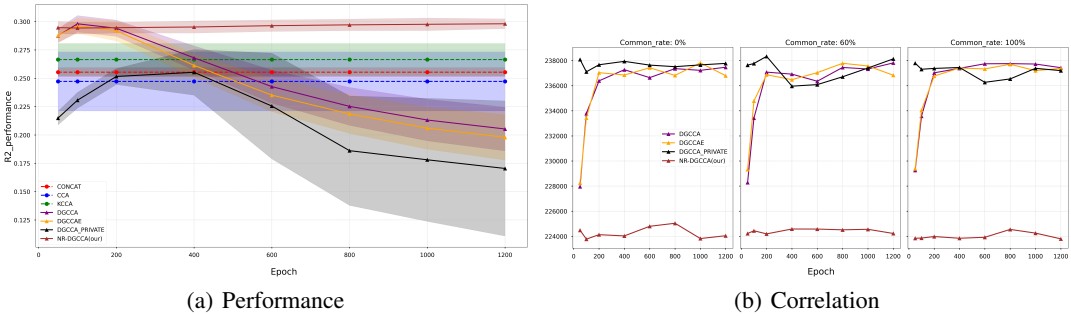

(a) Performance        (b) Correlation

Figure 10: (a) Mean and standard deviation of the GCCA-based method performance across synthetic datasets in different training epochs. (b) The correlation between noise and real data after transformation varies with epochs in different common rate settings for GCCA-based methods.

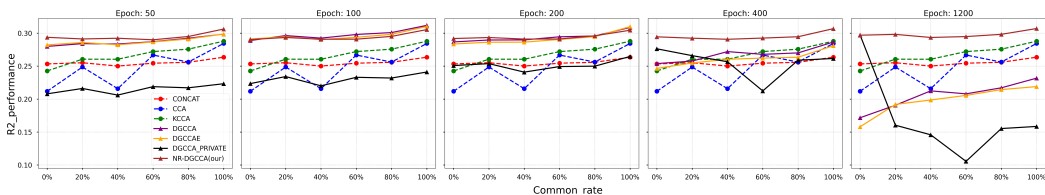

Figure 11: Performance of DGCCA-based methods with respect to different common rates during the training. Each column represents the testing accuracy of the method at a specific training epoch.

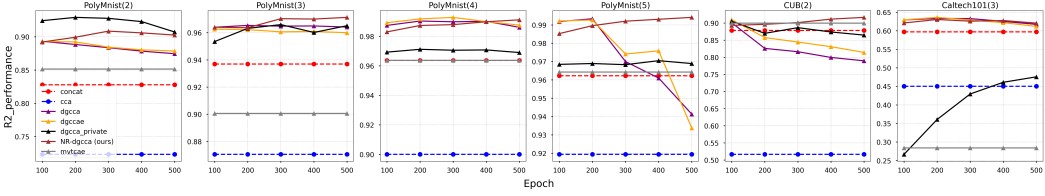

Figure 12: Performance of DGCCA-based methods in real-world datasets. Each column represents the performance on a specific dataset. The number of views in the dataset is denoted in the parentheses next to the dataset name.

### A.11 Additional experimental results

Table 2 and 3 present the model performance of various MVRL methods in synthetic and real-world datasets, and both tables correspond to the final epoch of the results presented in Figure 5 and 6. It should be noted that the values in Table 2 represent the mean and standard deviation of the methods across different tasks, indicating their performance and variability.

Table 2: Performance in synthetic datasets.

| R2/Common Rate | 0% | 20% | 40% | 60% | 80% | 100% |
|---|---|---|---|---|---|---|
| CONCAT | 0.253±0.038 | 0.255±0.039 | 0.250±0.040 | 0.254±0.040 | 0.256 ±0.042 | 0.264 ± 0.033 |
| CCA | 0.212±0.053 | 0.249±0.046 | 0.216±0.055 | 0.267±0.046 | 0.256±0.052 | 0.284±0.039 |
| KCCA | 0.243±0.047 | 0.261±0.046 | 0.260±0.043 | 0.272±0.045 | 0.276±0.049 | 0.288±0.038 |
| PRCCA | 0.212±0.053 | 0.249±0.046 | 0.216±0.055 | 0.267±0.046 | 0.256±0.052 | 0.284±0.039 |
| MVTCAE | 0.065±0.015 | 0.071±0.016 | 0.067±0.016 | 0.069±0.016 | 0.071±0.016 | 0.069±0.015 |
| DCCA | 0.136±0.036 | 0.188±0.040 | 0.194±0.044 | 0.192±0.049 | 0.215±0.044 | 0.221±0.036 |
| DCCAE | 0.134±0.056 | 0.200±0.043 | 0.211±0.040 | 0.224±0.042 | 0.228±0.043 | 0.230 ±0.040 |
| DCCA_PRIVATE | 0.279±0.044 | 0.143±0.043 | 0.14±0.042 | 0.114±0.04 | 0.139±0.041 | 0.144±0.042 |
| NR-DCCA (ours) | **0.296**±0.042 | **0.295**±0.045 | **0.293**±0.042 | **0.295**±0.045 | **0.300**±0.048 | **0.308**±0.038 |
| DGCCA | 0.172±0.039 | 0.191±0.044 | 0.212±0.039 | 0.208±0.042 | 0.217±0.042 | 0.232±0.04 |
| DGCCAE | 0.158±0.04 | 0.192±0.041 | 0.199±0.04 | 0.206±0.041 | 0.214±0.041 | 0.219±0.038 |
| DGCCA_PRIVATE | 0.297±0.043 | 0.16±0.045 | 0.146±0.038 | 0.106±0.044 | 0.155±0.041 | 0.159±0.035 |
| NR-DGCCA (ours) | **0.297**±0.043 | **0.298**±0.046 | **0.293**±0.043 | **0.295**±0.043 | **0.298**±0.048 | **0.307**±0.039 |

Table 3: Performance in real-world datasets

| F1 Score/Data | PolyMnist (2) | PolyMnist (3) | PolyMnist (4) | PolyMnist (5) | CUB | Caltech101 |
|---|---|---|---|---|---|---|
| CONCAT | 0.828 | 0.937 | 0.964 | 0.962 | 0.878 | 0.597 |
| CCA | 0.723 | 0.871 | 0.900 | 0.920 | 0.517 | 0.450 |
| KCCA | - | - | - | - | - | - |
| PRCCA | 0.712 | 0.849 | 0.899 | 0.918 | - | - |
| MVTCAE | 0.852 | 0.901 | 0.964 | 0.964 | 0.900 | 0.284 |
| DCCA | 0.870 | 0.959 | 0.975 | 0.934 | 0.805 | 0.604 |
| DCCAE | 0.871 | 0.958 | 0.983 | 0.965 | 0.850 | 0.605 |
| DCCA_PRIVATE | **0.923** | 0.963 | 0.972 | 0.969 | 0.853 | 0.480 |
| NR-DCCA (ours) | 0.913 | **0.969** | **0.991** | **0.993** | **0.921** | **0.625** |
| DGCCA | 0.875 | 0.964 | 0.986 | 0.941 | 0.790 | 0.617 |
| DGCCAE | 0.879 | 0.960 | 0.988 | 0.934 | 0.814 | 0.612 |
| DGCCA_PRIVATE | **0.907** | 0.965 | 0.969 | 0.969 | 0.864 | 0.476 |
| NR-DGCCA(ours) | 0.903 | **0.971** | **0.991** | **0.994** | **0.917** | **0.621** |

