# OpenReview forum: "Preventing Model Collapse in Deep Canonical Correlation Analysis by Noise Regularization"
_ICLR.cc/2024/Conference — Submitted to ICLR 2024_

### Official Review · Reviewer_hTsZ · 2023-11-01

**Soundness:** 4 excellent
**Presentation:** 4 excellent
**Contribution:** 4 excellent
**Rating:** 10
**Confidence:** 4

**Summary:**

The paper deal with Multi-View Representation Learning and solves a problem which affects Deep Canonical Correlation Analysis (DCCA) approaches, which is the model collapse. Model Collapse occurs when the performance of DCCA-based methods drops with the advancements of the training epochs, and is due to the model correlation among neural networks.
The authors make a comparison with simple CCA (not deep) approaches, and demonstrate that the main reason for CCA not having the model collapse issue is that the full-rank property holds in its transformation matrix, while the DNNs in DCCA do not possess such property. In particular, CCA searches for as a full-rank matrix, and it is robust to random noise, given the fact that the correlations before and after the (linear ) projection is kept
Therefore, the authors propose a noise regularization to enforce the DNNs to be “full-rank”, tailored for DCCA-based methods, dubbed NR-DCCA. NR-DCCA generates a set of i.i.d Gaussian white noise, with the same shape as the multi-view data Xk. Subsequently, DCCA is enforced to be full rank by adding a Noise regularized loss which requires that the correlation between the raw data and the noise is kept after having embedded the data into a latent space with the deep network, for each of the view.
Experiments on synthetic and real world datasets, against a set of SOTA approaches, show the validity of the simple idea of the authors

**Strengths:**

The problem is important, since DCCA-based MVRL methods without model collapse are hard to achieve.
A theoretical analysis of the model collapse issue in DCCA-based methods for MVRL is shown
A novel noise regularization approach is proposed for DCCA
The full-rank property of the CCA method which prevents model collapse is demonstrated, this drives the noise regularization approach from a theoretical perspective.
Experiments demonstrate a good performance. The comparative approaches are appropriate. The resulst tel a clear story, and there are no question marks that arise.

**Weaknesses:**

I did not find any major lack. I was focusing on the theory, but it seems very clear (yet simple). I was looking for additional, more appealing comparative approaches doing DCCA, but I did not find any. One may argue why the approach has been casted solely for Multi-View Representation Learning, since it could have a broader scope, but this is not a minus. Just a curiosity.
In general I think that the paper can be squeezed a little, in order to host some of the experiments of the additional material. In particular, I found fascinating the experiment reported in Fig.12, about the the correlation between unrelated data. This is a further proof of the goodness of the idea. I would also report the tsne visualization in the main paper.

**Questions:**

See my suggestions and curiosity above

---

> ### Author Response · Authors · 2023-11-17
> **Official Response to reviewer hTsZ**
>
> We really appreciate your recognition and positive feedback of our work! The itemized responses to your questions are as follows:
>
> > One may argue why the approach has been casted solely for Multi-View Representation Learning, since it could have a broader scope, but this is not a minus.
>
> Thanks for the thoughtful comments, and we believe this comment echoes Comment 3 by Reviewer 5U9E. The proposed Noise Regularization (NR) is indeed a way to regularize the neural network, and in general, it can be widely useful for different tasks.
>
> The developed “full-rank” property of the neural network stems from the full-rank of the transformation W_k, so this paper mainly focuses on associating DCCA and CCA. Current theoretical derivations only apply to CCA, and it is still challenging to provide a good explanation of why such NR would work in other tasks. This also echoes Comment 3 by Reviewer w9Bv and Comments 2 and 3 by Reviewer w9Bv. We do think this is an interesting area worth further exploration, and we will leave it for future study.
>
> > In general I think that the paper can be squeezed a little, in order to host some of the experiments of the additional material. In particular, I found fascinating the experiment reported in Fig.12, about the the correlation between unrelated data. This is a further proof of the goodness of the idea. I would also report the tsne visualization in the main paper.
>
> Thanks for your kind suggestions. We have squeezed the paper and merged the duplicate original Figures 2 and  5. We have also shown and analyzed the correlation of unrelated data in the revised section 5.2. We have mentioned the T-SNE results could also be evidence of model collapse in revised section 3.4 and pointed the readers to Appendix A.9 for details. We will further re-organize the paper to highlight the points the reviewer mentioned if it is accepted.
>
> Overall, we sincerely appreciate your positive feedback and hope that our responses have sufficiently answered your questions. Should there be any queries or points of discussion, we stand ready to provide further clarifications.

---

### Official Review · Reviewer_5U9E · 2023-11-03

**Soundness:** 3 good
**Presentation:** 3 good
**Contribution:** 3 good
**Rating:** 6
**Confidence:** 2

**Summary:**

The authors of this study empirically observe a critical issue of model collapse in DCCA-based methods. This phenomenon is characterized by a significant drop in performance as the training progresses. The model collapse issue poses a substantial challenge to the widespread adoption of DCCA-based methods, as it becomes difficult to determine the appropriate stopping point during training. To address this issue, the authors introduce NR-DCCA, which incorporates a novel noise regularization approach designed to prevent model collapse. Furthermore, they provide theoretical insights demonstrating that maintaining the full-rank property is essential for preventing model collapse, and the proposed noise regularization effectively enforces this property within the neural network. Additionally, they develop a framework for generating synthetic data containing various common and complementary information to facilitate a comprehensive comparison of Multiple View Representation Learning (MVRL) methods.

**Strengths:**

NR-DCCA consistently outperforms the baseline methods in both synthetic and real-world datasets. Moreover, the proposed noise regularization approach can be applied to other DCCA-based methods, such as DGCCA. This paper is well-structured and easy to follow.

**Weaknesses:**

Some key details are missing. See my comments below.

**Questions:**

However, the reviewer has several questions:

The proposed method involves adding a regularization term to the loss function. The choice of the optimal value for the hyperparameter α in Equation 6 is crucial. If α is set too large, the model may tend to behave like an identity mapping function, potentially reducing its effectiveness. Conversely, if α is too small, the model may not maintain a "full-rank" property. The authors should provide guidance or suggestions on how to select an appropriate α value.

The reviewer is interested in whether there is any theoretical analysis of the generalization ability of the proposed method. An exploration of how NR-DCCA's performance might extend to new, unseen data or domains would add depth to the paper.

The paper showcases the performance improvement of the proposed method in downstream tasks using off-the-shelf methods like Support Vector Regression (SVR). However, the reviewer is curious about whether there are any significant performance differences when fine-tuning the system using downstream tasks. An analysis of how NR-DCCA performs in this scenario would be informative.

---

> ### Author Response · Authors · 2023-11-17
> **Official Response  (1/2)  to reviewer 5U9E**
>
> We appreciate your comments and feedback. In addition to the general response, we address your itemized concerns here.
>
> > The proposed method involves adding a regularization term to the loss function. The choice of the optimal value for the hyperparameter α in Equation 6 is crucial. If α is set too large, the model may tend to behave like an identity mapping function, potentially reducing its effectiveness. Conversely, if α is too small, the model may not maintain a "full-rank" property. The authors should provide guidance or suggestions on how to select an appropriate α value.
>
> Thanks for the thoughtful comment. Yes, the choice of the hyperparameter α is essential in NR-DCCA. However, different from the conventional hyperparameter tuning, the determination of α is actually simpler, as we just need to search for the smallest α that can prevent the model collapse, and the model collapse can be directly observed on the validation data.
>
> In  Appendix A.6.2, we present performance curves of NR-DCCA in CUB under different α. if α is too large, the performance will converge slowly; if it is too small, the model collapse issue may remain. We increase the α adaptively until the model collapse issue is tackled, i.e., the correlation with noise will not increase or the performance of DCCA will not drop when the training progresses. The optimal α values of NR-DCCA for the CUB, PolyMnist, and Caltech datasets are found to be 1.5, 5, and 15, respectively.  Additionally, one can see the NR-DCCA outperforms DCCA robustly with a wide range of α.
>
> > The reviewer is interested in whether there is any theoretical analysis of the generalization ability of the proposed method. An exploration of how NR-DCCA's performance might extend to new, unseen data or domains would add depth to the paper.
>
> Thanks for the comment. The proposed Noise Regularization (NR) is indeed a way to regularize the neural network, and in general, it is widely acknowledged that regularization on neural networks is helpful for generalization ability [1,2].
>
> The common regularization methods include two categories: explicit methods (i.e. regularizing the parameters, hidden features, or gradients in the neural networks) [1]  and implicit methods (e.g., noise regularization) [2]. It is clear that the proposed NR falls in the second category, which lacks comprehensive studies in representation learning.
>
> Actually, the implicit methods usually “regularize the behavior” of neural networks. We believe this is a new perspective and our numerical experiments also indicate its outperformance. However, it might be challenging to analyze what has been regularized in neural networks, and hence it would be difficult to provide theoretical analysis regarding the generalization ability, due to the intricate neural network structures. Here we actually observe a trade-off between interpretability and performance in regularization methods (explicit vs. implicit). Instead, what we did in the paper is to show the analogy with the classic CCA, and we let the neural network mimic the good behaviors of CCA, given the fact that CCA has good generalization ability.
>
> We appreciate that the reviewer pointed out the right direction for potential future work. The primary objective of this paper lies in addressing the model collapse issue within DCCA, while we believe analyzing the detailed properties of such NR would be impactful.
>
> [1] Huang, Lei, et al. "Normalization techniques in training dnns: Methodology, analysis and application."
>
> [2] Poole, Ben, Jascha Sohl-Dickstein, and Surya Ganguli. "Analyzing noise in autoencoders and deep networks." arXiv preprint arXiv:1406.1831 (2014).

---

> ### Author Response · Authors · 2023-11-17
> **Official Response (2/2) to reviewer 5U9E**
>
> > The paper showcases the performance improvement of the proposed method in downstream tasks using off-the-shelf methods like Support Vector Regression (SVR). However, the reviewer is curious about whether there are any significant performance differences when fine-tuning the system using downstream tasks. An analysis of how NR-DCCA performs in this scenario would be informative.
>
> Thanks for the thoughtful comment. It is possible to also fine-tune the representation learning during the training of the downstream tasks (like SFT in LLM), while this approach deviates from the standard validation procedure of multi-view representation learning (MVRL).
>
> In MVRL, the predominant validation procedure is first to generate representations, and then use the representation directly. To ensure a fair comparison with existing methods, we still follow this procedure.
>
> Indeed, we believe that if we further fine-tune the representation for specific tasks, then the performance will definitely improve, but the representation will lose generality.
>
> This actually resembles other supervised multi-view problems, such as multi-view classification [3]. Investigating how NR-DCCA performs under these problems might be interesting and it is left for future work.
>
>
> Overall, we sincerely appreciate your insightful feedback and hope that our responses have sufficiently addressed any concerns raised. Should there be any queries or points of discussion, we stand ready to provide further clarifications.
>
> [3] Han, Zongbo, et al. "Trusted multi-view classification." arXiv preprint arXiv:2102.02051 (2021).

---

### Official Review · Reviewer_w9Bv · 2023-11-08

**Soundness:** 2 fair
**Presentation:** 2 fair
**Contribution:** 2 fair
**Rating:** 3
**Confidence:** 4

**Summary:**

This paper presents an algorithm for regularizing the multi-view deep CCA model with noise. Specifically, the algorithm introduces additional loss terms that encourage the correlation between DNN output of signal and noise, to be consistent with linear mapping output of signal and noise.

**Strengths:**

I like that the authors somewhat carefully designed numerical simulations to test their method.

**Weaknesses:**

I have several concerns.

1. Important references and discussions are missing. The fact that powerful neural networks could lead to degenerate feature space is known. In the original deep CCA paper [Andrew et al, 2013] there was already ridge regularization of the auto-covariance matrices (see their eqn 5) to avoid degeneracy and improve numerical stability. And that regularization technique was already studied for linear CCA, see
- De Bie and De Moor. On the regularization of canonical correlation analysis, 2003.
and an probabilistic interpretation where a Gaussian observation model leads to regularized covariance matrices
- Bach and Jordan. A Probabilistic Interpretation of Canonical Correlation Analysis. 2005.
While the authors' proposal appears different, it is still important to discuss the connections to existing methods and compare with them, both theoretically and empirically.

2. The intuition behind the proposal is not super clear to me. If the goal is to have full-rank f(X), so that the covariances are better conditioned, the abovementioned covariance regularization approach already achieves the same effect. If I look at Proposition 4, it essentially says CCA is invariant to linear transformations of input; but this is well-known and easy to see from the original formulation, even without complicated linear algebra. The more interesting analysis would be to explain why the correlation between signal and random noise is a good quantity for deep neural networks to mimic; there must be more structure than saying that fk is full-rank in my opinion.

3. The paper is not purely about optimization and numerical stability. And I expect to see different inductive bias from the proposed regularization. There shall be investigation of the feature quality on real-world datasets, as shown by prior deep CCA-based papers.

**Questions:**

- I hope to see non-trivial analysis regarding the effect of noise regularization.
- I hope to see comparison of feature quality against alternatives.

---

> ### Author Response · Authors · 2023-11-17
> **Official Response (1/3) to reviewer w9Bv**
>
> We appreciate your comments and feedback. In addition to the general response, we address your itemized concerns here.
>
> > Important references and discussions are missing. The fact that powerful neural networks could lead to degenerate feature space is known. In the original deep CCA paper [Andrew et al, 2013] there was already ridge regularization of the auto-covariance matrices (see their eqn 5) to avoid degeneracy and improve numerical stability. And that regularization technique was already studied for linear CCA...
>
> Thanks for the thoughtful comment. The reviewer mentioned the ridge regularization, which is a widely used technique to regularize the features (W_kX_k in CCA and f_k(X_k) in DCCA using our notations), and it is a default setting across almost all (D)CCA methods. In our experiments, **we have actually included ridge regularization during hyper-parameter tuning for all the methods with and without noise regularization**.
>
> However, the ridge regularization **mainly regularizes the features (i.e., W_kX_k in CCA and f_k(X_k) in DCCA), rather than the transformations (i.e., W_k in CCA and f_k in DCCA)**. W_k in CCA does not need any extra regularization as it is automatically full-rank, while the f_k is still unconstrained and unregulated in DCCA. Therefore, our study mainly focuses on the regularization of the transformation in DCCA. That is why we did not compare with the ridge regularization in the paper.
>
> The reviewer agreed that “powerful neural networks could lead to degenerate feature space”. However, the degenerated features produced by neural networks in DCCA can still be full-rank, and simply enforcing the covariance matrix to be full-rank by adding ridge regularization is not even helping, because the collapse of neural networks is intricate. Therefore, we develop the concept of the “full-rank” property of f_k (note this is an analogy of the full-rank property of W_k), and it is different from the full-rank of the features f_k(X_k). We apologize for the potential confusion raised by this definition.
>
> Overall, we believe that ridge regularization is mainly for combating noise and improving numerical stability, it cannot prevent the neural networks from degenerating (i.e., model collapse). In contrast, we should directly study the “full-rank” property of the neural networks f_k and we believe the relevant studies are lacking.
>
> To further support our arguments, we provide the experimental results with different ridge parameters on a real-world dataset CUB at 500-th epoch as follows:
>
> | Ridge parameter | Performance | Square sum of feature covariance |
> |-----------------|-------------|---------------------------------|
> | 0               | 0.805       | 0.019                           |
> | 1e-5            | 0.786       | 0.028                           |
> | 1e-4            | 0.776       | 0.210                           |
> | 1e-3            | 0.731       | 7.336                           |
>
> One can see that the ridge regularization **even damages the performance** of DCCA. In our NR-DCCA, we actually set the ridge parameter to **zero**. We conjecture the reason is that the large ridge parameter could make the neural network even “lazier” to actively project the data into a better feature space, as the full-rank property of features and covariance matrix are already guaranteed, and this is also evidenced by the “Square sum of feature covariance” shown in the table.
>
> We apologize that we should have included the above discussion in the paper, and we have now incorporated the above results in Appendix A.6.1 for clarification.

---

> ### Author Response · Authors · 2023-11-17
> **Official Response (2/3) to reviewer w9Bv**
>
> > The intuition behind the proposal is not super clear to me. If the goal is to have full-rank f(X), so that the covariances are better conditioned, the abovementioned covariance regularization approach already achieves the same effect. If I look at Proposition 4, it essentially says CCA is invariant to linear transformations of input; but this is well-known and easy to see from the original formulation, even without complicated linear algebra. The more interesting analysis would be to explain why the correlation between signal and random noise is a good quantity for deep neural networks to mimic; there must be more structure than saying that fk is full-rank in my opinion.
>
> We apologize for the potential misunderstanding caused, and we have updated the paper to make it clearer (in Section 3.4). Actually, **we define the “full-rank” property of f_k and it is different from the full-rank property of features  f_k(X_k)**. In CCA, the two concepts are nearly the same, as features W_kX_k are forced to be full-rank by the loss function, and W_k is naturally full-rank (refer to Lemma 2 in our paper). However, it is difficult to define what is a “full-rank” property of neural networks since it is not a matirx. Because the current DCCA has no regularization on the neural networks, we believe **such inconsistency is the potential reason for the model collapse in DCCA**.
>
> We agree with the reviewer that the Proposition looks straightforward. To be precise, the correlation with noise is invariant to full-rank linear transformation. However, it is the first time that the connection between the correlation with noise and the full-rank property of W_k is built. In terms of DCCA, as it is difficult to directly define the “full-rank” property of f_k, we first find the equivalent condition of the full-rank property of W_k in CCA, and then we can transplant this condition to DCCA. Specifically, because of Theorem 1 (Proposition 3 + Proposition 4), we can equivalently define the “full-rank” property of  f_k based on the noise in Definition 2.
>
> We do agree with that reviewer that more structure of the neural networks should be explored to further understand what has been regularized by our proposed Noise Regularization. However, we have to admit that interpreting the parameters in neural networks is challenging. Our way to tackle this problem is to look at the behavior of the neural networks. Through numerical experiments, we find that the CCA never collapses and the DCCA always collapses. Then we design the experiments more extremely to let both CCA and DCCA learn the correlation between two independent random noises. The results indicate that CCA can identify the invariant correlation between the noise, while DCCA cannot (we have moved related experiments into Section 5.2 ). We believe this is mainly because **the powerful neural networks have “created” correlation itself**. Therefore, we enforce the neural networks to preserve the correlation for random noise in DCCA, and this motivates our idea of noise regularization.
>
> **We call f_k “full-rank” to help the reader associate with the full-rank property of W_k, as the full-rank transformation matrix would not “create” correlation**. But this name might cause the reviewer’s misunderstanding. We apologize and we have incorporated more discussions to elaborate on the “full-rank” property of neural networks in the updated paper to eliminate confusion (in Section 4.2).

---

> ### Author Response · Authors · 2023-11-17
> **Official Response (3/3) to reviewer w9Bv**
>
> > The paper is not purely about optimization and numerical stability. And I expect to see different inductive bias from the proposed regularization. There shall be investigation of the feature quality on real-world datasets, as shown by prior deep CCA-based papers.
>
> Thanks for the comment. Indeed, our work is not about optimization and numerical stability. We **focus on imposing the inductive bias on the neural networks for DCCA**. The correlation with random noise is invariant to full-rank linear transformation, and hence we hope the correlation is also invariant to “full-rank” non-linear transformation. Hence we define such “full-rank” property as the inductive bias of the neural networks.
>
> In terms of the investigation of the **feature quality** on real-world datasets, our evaluation of feature quality is mainly guided by **visualization techniques** (as demonstrated by TSNE in Appendix A.9 ). Additionally, the **performances** across a multitude of downstream tasks, encompassing both synthetic and real-world datasets, demonstrate the consistent outperformance of the proposed method. Moreover, guided by our theory, we reported the **correlation between the outputs of unrelated data**, which provides another way to observe feature quality and model collapse issues.
>
>
> > I hope to see non-trivial analysis regarding the effect of noise regularization.
>
> Please check our response to your Comments 1 and 2.
>
> > I hope to see comparison of feature quality against alternatives.
>
> Please check our response to Comment 3.
>
> Overall, we sincerely appreciate your insightful feedback and hope that our responses have sufficiently addressed any concerns raised. Should there be any queries or points of discussion, we stand ready to provide further clarifications. In light of the revisions and discussions, we kindly invite you to reconsider your score on this paper.

---

> ### Author Response · Authors · 2023-11-21
> **Looking for Further Discussion**
>
> Dear Reviewer w9Bv,
>
> We are aware that after tomorrow, the opportunity for further discussion will no longer be available in this year's ICLR.  We wonder if we have adequately addressed the concerns and clarified the misunderstanding you previously raised?
>
> It is our earnest hope to continue our discussion, should there be any lingering questions or points needing clarification.

---

> ### Comment · Reviewer_w9Bv · 2023-11-22
> **Reply**
>
> I appreciate the authors providing details on the tuning of ridge parameters for deep CCA. I am OK on that aspect.
>
> I understand at the high level that the intuition is to not create non-existent or spurious correlation between the views, when we use powerful neural networks for parameterizing feature mappings. But still
> - I feel the desired property is more than the "full-rank" property proposed in this paper, and this property has to be characterized more concretely (for example, similarly to how adding noise to data is shown to be equivalent to L2 regularization under assumptions). This is important since the authors claim their method to be advantageous over ridge regularized DCCA, which adds isotropic noise to data to ensure covariance is well-conditioned.
> - It is not very clear how the additional loss on matching correlation obtain with DNNs with linear mappings achieves the desired property. In fact, because noise is independently generated from data, the true correlation between noise and data should be zero (with infinite data). Why not instead minimize correlation between data and noise for regularizerization?

---

> > ### Author Response · Authors · 2023-11-23
> > **Further Response (2/2) to Reviewer w9Bv**
> >
> > >It is not very clear how the additional loss on matching correlation obtain with DNNs with linear mappings achieves the desired property. In fact, because noise is independently generated from data, the true correlation between noise and data should be zero (with infinite data). Why not instead minimize correlation between data and noise for regularizerization?
> >
> > As we discussed and agreed before, correlation with noise should be irrelevant to full-rank linear transformations and hence it is natural that we hope neural networks in DCCA hold the same property. So we do not minimize the correlation with noise but maintain it unchanged.
> >
> > Indeed, when the batch size is infinite, the correlation between the original input and noise should be zero. We actually tested this regularization term during the study. It is found that when the batch size is 20000 and the feature dimension is 100, the Corr(X_k, A_k) is close to zero.
> >
> > However, in practical use,  the size of the train set may be very small, especially during batch training. Hence removing Corr(X_k, A_k) may cause instability and minimize abs(Corr(f_k(X_k), f_k(A_k))-Corr(X_k, A_k)) is more “soft” in terms of numerical stability.
> >
> > In the CUB dataset, there is only 480 training data. We try to directly minimize Corr(f_k(X_k), f_k(A_k)). We report the performances of the 500th epoch.
> >
> >
> > |  Methods | Performances |
> > |---------|---------|
> > | MVTCAE |0.900 |
> > | DCCA  | 0.827 |
> > | DCCAE | 0.850 |
> > | DCCA_PRIVATE | 0.853 |
> > | NR-DCCA(minimizing Corr(f_k(X_k), f_k(A_k)) ) | 0.839 |
> > | NR-DCCA(original)| 0.921 |
> >
> >
> > One can see that minimizing Corr(f_k(X_k), f_k(A_k)) is helpful for preventing model collapse. However, this regularization cannot reach the performance of our proposed NR, which is 0.921. Therefore, this regularization is not adopted in our paper.
> >
> > We sincerely appreciate your insightful feedback and hope that our responses have sufficiently addressed any concerns raised.

---

> ### Author Response · Authors · 2023-11-23
> **Further Response (1/2) to Reviewer w9Bv**
>
> >I appreciate the authors providing details on the tuning of ridge parameters for deep CCA. I am OK on that aspect.
> I understand at the high level that the intuition is to not create non-existent or spurious correlation between the views, when we use powerful neural networks for parameterizing feature mappings.
>
> We appreciate your feedback on our response and thoughtful comments. Great to know that the reviewer understands the key idea in this paper, and we also understand that the reviewer request a deeper theoretical understanding of the proposed noise regularization (NR) approach. We actually thought about this, but indeed the further exploration of such a “full-rank” property was non-trivial, because our NR approach is not simply adding noise to the input,  and f_k can be any form of neural networks. However, we still believe our NR approach is quite new, and it achieves state-of-the-art performance consistently, so we would like to share this approach with the community.
>
> >I feel the desired property is more than the "full-rank" property proposed in this paper, and this property has to be characterized more concretely (for example, similarly to how adding noise to data is shown to be equivalent to L2 regularization under assumptions). This is important since the authors claim their method to be advantageous over ridge regularized DCCA, which adds isotropic noise to data to ensure covariance is well-conditioned.
>
> Thanks a lot for the thoughtful comments! As the reviewer concluded, our desired property of neural networks is to “avoid creating non-existent correlation between views”, and this property is actually intricate because it is challenging to regularize the weights in neural networks to obtain this property. In contrast, this paper enforces the property directly, while it is indeed non-trivial to connect this property to the weights of neural networks.
>
> Our NR approach differs from the standard NR approach, as we did not add noise to the data directly and the standard NR approach requires strong assumptions (e.g., autoencoder structures) for analysis, so it is challenging to obtain similar properties like L2 regularization in our approach. As we mentioned earlier, we believe our NR approach mainly regularizes the behavior of the neural network, and how the parameters are regularized actually depends on the structure of the neural network, which we do not restrict directly.
>
> It might be possible to derive some analytical properties of our NR approach if we assume specific structures of the neural networks. For example, [1] assumes an autoencoder structure with a linear decoder, and [2] relies on the linear transformation in CCA. However, this paper aims to develop a generalized NR approach for deep CCA, so we did not touch on this aspect. We do think this will be a great direction for further exploration.
>
> Overall, we still believe our NR approach is quite new, as it provides a new aspect to regularize the neural network, and its performance is consistently state-of-the-art, so we would like to share this finding with the community. Thanks again for your thoughtful comments and for pointing out the potential directions for further investigation!
>
>
> [1] Poole, Ben, Jascha Sohl-Dickstein, and Surya Ganguli. "Analyzing noise in autoencoders and deep networks." arXiv preprint arXiv:1406.1831 (2014).
>
> [2] De Bie, Tijl, and Bart De Moor. "On the regularization of canonical correlation analysis." Int. Sympos. ICA and BSS (2003): 785-790.

---

### Official Review · Reviewer_ZnnT · 2023-11-28

**Soundness:** 2 fair
**Presentation:** 2 fair
**Contribution:** 1 poor
**Rating:** 3
**Confidence:** 5

**Summary:**

This paper proposes to deal with the “model collapse” problem of deep canonical correlation analysis. The technical approach is to introduce a noise regularization term.  Basically, the regularization asks that, in each view, the correlation with Gaussian noise before transforming the data and after transforming the data to be similar. Some properties of such regularization under the linear case is studied.

**Strengths:**

The reviewer does agree with the paper that the DCCA formulation itself does not guarantee meaningful representation learning, as the learned solution may lose information of the data.
However, this is a well recognized problem and was studied from many aspects in the literature (DCCA was from 2013 and some fixes were proposed in Wang et al 2015 already) - some of them have rigorous and interesting proofs, which this submission lacks.

**Weaknesses:**

**The mathematical rigor of this submission is questionable.** The so-called “model collapse” seems to have no rigorous definition. There is a mentioning “The correlation between unrelated data increases as the collapsed model transforms any data to a degenerated feature space” in the end of Section 3. This seems to imply that model collapse equals to finding a “degenerated feature space”, which also does not have a rigorous definition.

**The discussion/comparison with existing methods is lacking.** My understanding is that the noise regularization is a heuristic to make the learned representation have relatively large entropy - so that the solution is not “collapsed” or “degenerated”. This is a reasonable heuristic. The paper did not provide any other justifications other than intuition. Note that it is unclear why the proposed method is better at avoiding degenerated solutions compared with many existing methods, e.g., the DCCAE [wang et al 2015], i.e., deep CCA with an autoencoder to maintain the information from data. The paper mentioned that "... Wang et al. (2015) introduces the reconstruction errors of autoencoders to DCCA ... However, the model collapse issue of DCCA-based methods has not been explored and addressed". However, it is clear to me that the autoencoder can effectively avoid degenerate solutions. As there is no formal definition of "model collapse", it is unclear why DCCAE cannot avoid it.

**The key claimed theoretical contribution is questionable.** One of the key claimed contributions is “Rigorous proofs are provided to demonstrate that the full-rank property of the transformation in the CCA method is the key to preventing model collapse, which justifies the developed noise regularization approach from a theoretical perspective. ” There are a series of concerns regarding this claim and its related developments.

-  First, the major theorem (Theorem 1) is a little hard to comprehend. It says eta_k = 0 if and only if the square matrix W_k has full rank. It is not easy to follow why a square matrix is applied for CCA as CCA almost always uses a “fat matrix” W_k for dimensionality reduction. Hence, it is hard to understand how this theorem is useful in practice.

-  Second, more importantly, using the linear case to argue for nonlinear cases seems to be far-fetching. Note that even if X_k has zero-mean, there is no guarantee that f(X_k) still has zero mean. This basically breaks the proof of the linear case immediately. Saying that ``rigorous proof for linear cases’’ can be used to justify the nonlinear case is not convincing.

-  Third, it is hard to understand the relation between “full rank W_k” and “model collapse”, as these terms are never formally defined or linked together.

-  Fourth, Definition 2 seems to be arbitrary.

**Questions:**

I do not have questions for the authors.

---

### Author Response · Authors · 2023-11-17
**General Response to all Reviewers**

We thank all reviewers for their questions and constructive feedback. We have updated the paper and appendix, and the key changes in the revised submission include:

Revision 1: We rewrote section  3.4  to clarify that our paper aims to regularize the neural networks, denoted as f_k, to possess an intrinsic "full-rank" property. Simply forcing the features f_k(X_k)  to be “full-rank” is not enough to prevent model collapse in DCCA.

Revision 2: we incorporated more discussions to elaborate on how we associate the “full-rank” property of neural networks f_k with the full-rank property of W_k. We further clarified the difference between our full-rank property defined in correlation with noise and that defined in matrix rank in the revised Section 4.2.

Revision 3: We pointed out that the model collapse and feature quality in DCCA could be observed by decline-in-performance, correlation between unrelated data, and feature visualization in the revised section 3.4. Additionally, we added the analysis of the correlation between unrelated data into section 5.2.

Revision 4: We provided the experimental results with different ridge parameters on a real-world dataset CUB in Appendix A.6.1.

Revision 5: We added procedures to advise on how to choose the NR loss parameter in Appendix A.6.2.


We summarize a few key clarifications regarding the review comments and the changes above:

Response (1) to Reviewer w9Bv clarifies that ridge regularization is a common technique of CCA and we have actually included ridge regularization during hyper-parameter tuning for all the methods with and without noise regularization. Moreover, it does not prevent model collapse in DCCA. This prompted Revisions 1 and 4.

Response (2) to Reviewer w9Bv clarifies that the inconsistency of transformation processes between CCA and DCCA  is the reason for model collapse issues. Our Theorem 1 is straightforward yet profound, as it first time provides the equivalent condition of the full-rank property of W_k by connecting it to the correlation with noise, and then we can transplant this condition to DCCA. This prompted Revision 2.


Response (3) to Reviewer w9Bv clarifies that we hope the correlation with noise is invariant to the “full-rank” non-linear transformation f_k, which is consistent with CCA and hence we define such “full-rank” property as the inductive bias of the neural networks.  Additionally, we have further pointed out that the feature quality, as well as model collapse, could be observed by decline-in-performance, correlation between unrelated data, and feature visualization in revised section 3.4. This prompted Revisions 2 and 3.

Response (1) to Reviewer 5U9E clarifies that the determination of α is actually simple, as we just need to search for the smallest α that can prevent the model collapse and we have added an example in Appendix A.6.2. This prompted Revision 5.

Response (2) Reviewer hTsZ clarifies that we have realized it is important to show evidence of model collapse, including correlation between unrelated data and feature visualization, more directly in the main paper. This prompted the Revision 3.

By following the reviewers’ suggestions and comments, we believe we have further strengthened the manuscript.
Also, we really appreciate that reviewers have pointed out the future direction of this work:
Responses (2,3) to Reviewer 5U9E and Response (1) to hTsZ clarify that, in general, NR loss regularizes the behavior of neural networks and it might be useful in other tasks such as domain generalization and multi-view classification. However,  the developed “full-rank” property of the neural network stems from the full-rank of the transformation W_k, so this paper mainly focuses on associating DCCA and CCA. Current theoretical derivations only apply to CCA, and it is still challenging to provide a good explanation of why such NR would work in other tasks. We do think this is an interesting area worth further exploration, and we will leave it for future study.

---

### Meta-Review · Area_Chair_knSC · 2023-12-07

**Metareview:**

This paper proposes a regularization method to improve deep CCA features, by encouraging correlation between DNN output of signal and noise, to be consistent that of linear mapping output of signal and noise. While the idea seems plausible and the authors have shown some empirical evidence that the regularization is useful, there exists a large understanding gap between the theoretical analysis in the linear case and the practice for DNNs. Reviewers have suggested to more rigorously define "collapse" and study how the regularization mitigates it. Also the true correlation between data and noise shall be zero and the authors shall reason minimizing deep correlation in that case is worse than the proposed method.

**Justification For Why Not Higher Score:**

For a relatively simple method, the effect of regularization is not understood to the extent that the results are convincing.

**Justification For Why Not Lower Score:**

N/A

---

### Decision · Program_Chairs · 2024-01-16

Reject